# Exploring Sports Nutrition Knowledge in Elite Gaelic Footballers

**DOI:** 10.3390/nu13041081

**Published:** 2021-03-26

**Authors:** Luke O’Brien, Kieran Collins, Farzad Amirabdollhian

**Affiliations:** 1School of Health Sciences, Liverpool Hope University, Hope Park, Liverpool L16 9JD, UK; amirabf@hope.ac.uk; 2Gaelic Sports Research Centre, Technological University Dublin, Tallaght, D24 FKT9 Dublin, Ireland; kieran.collins@tudublin.ie

**Keywords:** Gaelic games, sport nutrition, questionnaire, nutrition knowledge

## Abstract

Nutrition intake plays a crucial role in improving athletic performance, enhancing adaptations to training, and augmenting recovery from exercise. However, research has reported that Gaelic footballers consistently fail to meet energy and carbohydrate recommendations. Sports nutrition knowledge (SNK) can influence the dietary intake of athletes, and therefore has the potential to have a significant impact on athletic performance. The aim of this study was to investigate the current level of SNK in elite Gaelic footballers (*n* = 100). An online version of the Nutrition for Sport Knowledge Questionnaire (NSKQ) was used to assess sports SNK. The overall mean SNK scores for Gaelic footballers and practitioners were 47.6 ± 12.3% and 78.1 ± 8.3%, respectively. There were no differences in knowledge between age groups, education level or divisional status. The top three sources of nutrition information identified by participants were team dietitian/nutritionists (84.0%), athletic trainers/strength and conditioning coaches (73%), and social media (37%). The results show that there is a major gap in the SNK of Gaelic footballers, while practitioners demonstrated a promising SNK, that could support Gaelic footballers. There is a need for development of interventions and knowledge transfer partnerships, including more effective methods of educating Gaelic footballers and translating sports nutrition principles to players. Developing appropriate nutritional education strategies using online resources and mobile applications could help to improve nutritional knowledge and practice of Gaelic footballers.

## 1. Introduction

Gaelic football is one of the national sports of Ireland, and it is a physically demanding intermittent high-intensity invasion team sport [1]. Elite players cover a total distance of between 8160 and 9222 m during competitive match play with up to 1731 ± 659 m of this completed at high speed (≥17 km·h^−1^) [2,3,4,5]. The physical nature of the game, large distances covered, and high sprint speeds displayed during elite match-play requires players to be competent in many aspects of fitness such as aerobic and anaerobic fitness, muscle strength, strength endurance, speed, and agility [6,7]. Despite its amateur status, preparation for training and competition are comparable to that of professional sport [8,9]. Due to the high-intensity nature of Gaelic football training and match play, meeting energy and nutrient needs in training and competition should be a high priority for athletes to maintain lean muscle mass, enhance recovery and improve performance [10].

Elite Gaelic footballers training programs vary depending on the phase of the season. During preseason, players can train up to 4–5 days per week with focus on strength development and field-based conditioning [7,9]. The main focus of the competitive season is peak performance and recovery, and the number and duration of training sessions during the week will decrease as game load increases [8,11]. Energy, carbohydrate (CHO), and protein requirements of athletes should be periodized alongside their training and competition program to support training adaptations, fuel optimal performance and augment recovery from competition [12]. Although there are no widely agreed nutritional recommendations produced specifically for Gaelic football, internationally endorsed sport nutrition guidelines appropriate for similar team sports suggest players should aim for a CHO intake of 6–10 g·kg^−1^ on training days to support 1–3 h·day^−1^ of moderate to high-intensity exercise [12]. Depending on training demands, CHO requirements may reduce to 3–5 g·kg^−1^ (low-intensity) or 5–7 g·kg^−1^ (moderate-intensity) [12]. Based on the training demands and energy expenditure of players, CHO intakes of 7 g·kg^−1^ can fuel training demands and achieve energy balance during the pre-season training period [9]. To promote optimal performance in competitive Gaelic football match play, 7–12 g·kg^−1^ should be consumed during the 24 h preceding [12]. Previous research has reported that Gaelic football players consistently fail to meet energy and CHO recommendations during a training week [9,13]. Likewise, CHO intakes of Gaelic footballers in the days preceding competition have been reported to be <4 g·kg^−1^ [14,15], which is below the recommended intakes [12].

Good nutritional practice among athletes is associated with improved performance, enhanced adaptations to training and augmented recovery post-exercise [16], however, previous studies demonstrated that Gaelic footballers have sub-optimal dietary intakes [9,14,15]. The underlying reasons for this are unclear, it may be due to lack of time, cooking skills, finance, attitude towards nutrition, or nutrition knowledge [17]. Sports nutrition knowledge (SNK) can influence athletes’ dietary intake, and in turn, impact athletic performance [18,19]. In volleyball players, improvements in nutrition knowledge were observed following a dietitian lead nutrition education intervention, corresponding with increased energy and CHO intake, in line with sport nutrition guidelines. Similarly, Rossi et al. (2017) reported that a sport nutrition education intervention in baseball players increased nutritional knowledge and nutritional status [20]. Furthermore, the sport nutrition education intervention resulted in a reduction body fat percentage and improved athletic performance [21]. Athletes that demonstrated higher nutrition knowledge consume more fruit, vegetables and CHO rich foods than those with lower levels of nutrition knowledge [22]. Previous research has identified that both elite and sub-elite Gaelic football players demonstrated poor nutrition knowledge [23,24,25]. Similarly, elite female Gaelic games players and elite male hurlers demonstrated inadequate levels of nutrition knowledge [26,27].

The recent attempts to examine the SNK of male [23,24,25] and female [26] Gaelic football players is addressing a gap in the body of knowledge. On the other hand, there are some methodological limitations within the previous studies that would warrant the current study as the logical continuation of the previous research. For instance, within the literature examining the SNK of male Gaelic football players, the study of Magee et al. (2016) investigated the nutrition knowledge of university and club level athletes, in relation with hydration before and after training. The inadequate discrimination between the wide variety of sports (e.g., golf, hockey, Gaelic football, netball, rugby, running and Shotokan karate) included within this study, interlinking and limiting the examination of SNK to hydration, and lack of discrimination between the level of athletes limited the internal validity of the findings [23]. Similarly, within the study of McCrink et al. (2020), despite a relatively large number of participants (*n* = 168) participating in the study of the dietary intake of Gaelic football players, the SNK was examined only in a small sub-sample of the study population (*n* = 24) limiting the external validity and the generalizability of the findings [24]. The study of Renard et al. (2021) including 68 club (sub-elite) and 84 inter-county players (elite) Gaelic football players, is the most recent and most focused examination of the SNK demonstrating a total score of 44.3 ± 12.7% (corresponding with poor nutrition knowledge) with no difference between playing levels. Nevertheless, a shorter form of the SNK questionnaire was used in this study focusing around athlete’s declarative nutrition knowledge (in comparison with the procedural and tacit knowledge), while particular methodology in sampling, restricted the ability to determine the response rate [25].

It is unclear as to why athletes often demonstrate poor nutrition knowledge [17], and inadequate access to reputable nutrition information has been one of the reasons postulated [28]. Preferred sources of nutrition information among athletes vary, as they may obtain information from a nutritionist/dietitian, strength and conditioning coach, the media and the internet, their peers, teammates and family, and their coach [28]. Recently, Renard et al. (2020) reported that female Gaelic games players most preferred nutrition information source was a nutritionist; however, only 20% reported having access to nutrition information and 16% had access to a nutritionist [26]. 

The number of sports nutrition practitioners working with team sport athletes and providing nutrition advice has increased [29]. Devlin and Belski (2016) reported that 97.8% of elite Australian football players had access to a sports dietitian and choose them as their main source of information [30]. There is limited information regarding Gaelic footballer’s access to relevant and appropriate nutrition advice. It is imperative that sports nutrition practitioners be suitably qualified, have a comprehensive understanding of evidence-based sport nutrition recommendations and support athletes by improving compliance with sports nutrition guidelines [31].

The aim of this study was to evaluate the SNK of Gaelic footballers. Identifying knowledge gaps would enable sports nutrition practitioners to target and implement nutrition education interventions to improve nutritional practices of Gaelic footballers. 

## 2. Materials and Methods

### 2.1. Experimental Design and Participants 

This cross-sectional study administered a SNK questionnaire to elite Gaelic football players competing in the All-Ireland senior football championship and also to sports nutrition practitioners’ working with elite Gaelic football squads. Eighteen inter-county Gaelic football teams were recruited via their team nutritionist/athletic trainers and were contacted via email and asked to forward a link to the questionnaire to athletes in their squad. Practitioners were also asked to complete the questionnaire themselves if they provided nutrition advice to players. The inclusion criteria included inter-county player currently part of an inter-county squad or a practitioner currently providing advice to inter-county players. One hundred players and eight practitioners completed the questionnaire. For this study, participants enrolled in teams from all divisions were considered as elite athletes. Participants were provided with a participant information sheet and online consent form and agreed to participate electronically. The study protocol was approved by the local ethics committee of Liverpool Hope University (Ethics code: S 07-06-2018 PA 030).

### 2.2. Procedures

Elite Gaelic footballers were invited to participate via direct recruitment from inter-county sports nutrition practitioners or athletic trainers. E-mails which contained a link to the online questionnaire were sent to practitioners involved within elite Gaelic football squads. They were asked to forward the email with the link to the questionnaires and instructions for completion to their athletes. The questionnaire was uploaded onto a secure clinical research online survey platform (Bristol Online Surveys, University of Bristol). More information on the study, along with the consent form, was provided on the first page of the questionnaire.

### 2.3. Instruments

SNK was assessed using Nutrition for Sport Knowledge Questionnaire (NSKQ). This questionnaire was deemed appropriate to use with adult male Gaelic football players as it had previously been used to assess SNK in adult male team sport athletes [32]. The NSKQ was robustly validated in Australian athletes [33]. Content, face and construct validity were tested during the questionnaire’s development and the questionnaire demonstrate excellent test–retest reliability and internal consistency reliability [33]. The questionnaire included demographic questions on age, education, competition level and sources of nutrition information. The questionnaire included 89 questions across six sub-sections: weight management (*n* = 13), macronutrients (*n* = 30), micronutrients (*n* = 13), sports nutrition (*n* = 13), supplementation (*n* = 12) and alcohol (*n* = 8). The sports nutrition section includes items on hydration (*n* = 4) and nutrition before (*n* = 1), during (*n* = 6) and after (*n* = 2) training and competition. Correct answers were given + 1 score, while incorrect answers or “unsure” were given a score of 0. All correct answers were totalled and resulted in the NSKQ score for each participant, for each sub-section, and for the whole questionnaire, which was converted to percentage. Further, players were also asked to list where they obtained nutritional information (i.e., dietitians, coaches, teammates, Internet, etc.). During the validation of the questionnaire, individuals with nutrition education achieved scientifically greater mean scores than individuals without a formal nutrition education, which indicates good construct validity [33]. Scores can be classified by the following: “poor” (0–49%), “average” (50–65%), “good” (66–75%) and “excellent” knowledge (76–100%) [33]. These classifications were based on the known-group comparison analysis, indicative of the construct validity within the original validation of the SNKQ [33]. The questionnaire was delivered online via the Bristol online survey tool. 

### 2.4. Statistical Analysis

Descriptive statistics for demographic data and knowledge scores are presented as mean ± standard deviation. In this study ‘sport nutrition knowledge score’ refers to the proportion (%) of the correct answer. All statistical analyses were conducted in SPSS (IBM Corp. Released 2016. IBM SPSS Statistics for Windows, Version 24.0. Armonk, NY, USA: IBM Corp). The normal distribution of variables was assessed using the Shapiro–Wilk test. Differences in knowledge scores based on age, competition level and highest level of education were assessed using a one-way ANOVA. Where ANOVA results were significant, a Bonferroni post hoc analysis was conducted to determine which groups differed. *p*-values for significance testing were set at *p* ≤ 0.05.

## 3. Results

### 3.1. Participant Characteristics

Eight inter-county team athletic trainers/sports nutrition practitioners agreed to forward the questionnaire to their players. One hundred and eighty-five male players started the questionnaire. After deleting incomplete responses, there were 100 productive questionnaires deemed usable for inclusion in the analysis. The mean age (±SD) of all athletes was 25.7 ± 3.7 years. As the majority (86%) of the Gaelic football adult players who participated in this study were young adults aged below 30 years, the age groups were categorized as 18–23 years, 24–29 years and aged 30 years and over to clarify the demographic. Participants who completed the questionnaire competed in the National football league division 1 (43%), 2 (25%), 3 (15%) and 4 (17%). Participant characteristics of the Gaelic footballers who completed the NSKQ are presented in Table 1.

### 3.2. Sports Nutrition Knowledge Scores

The overall mean SNK score for Gaelic footballers and practitioners were 47.6 ± 12.3% and 78.1 ± 8.3%, respectively. Only 7% of Gaelic footballers achieved an above average SNK score of 65% or higher. Meanwhile, 100% of practitioners attained an above average SNK score of 65% or higher, while 62.5% of practitioners scored an excellent SNK score of 75% or higher. Scores (mean ± SD; %) for each subsection of the NSKQ, are presented in Table 2.

### 3.3. Sport Nutrition Knowledge among Subgroups

A one-way ANOVA was used to compare the mean scores of the NSKQ between different age groups, educational levels and divisional status but no significance was found (Table 3).

### 3.4. Nutrition Information Sources

The top three sources of nutrition information identified by participants were team dietitian/nutritionists (84.0%), athletic trainers/strength and conditioning coaches (73%) and social media (37%) (Table 4).

## 4. Discussion

The aim of this study was to investigate the SNK among elite male Gaelic football players. The overall SNK of players was poor (47.6%), and this is consistent with previous research on Gaelic games athletes [23,24,25,26,27], direct comparison of nutrition knowledge scores between studies can be difficult due to the use of different assessment tools. However, similar knowledge scores were observed for both female and male Gaelic footballers using an abridged version of the same questionnaire [23,26], while sub-elite Gaelic footballers reported lower levels of nutrition knowledge using the same question [24]. The difference between elite and non-elite players could be explained by the fact that the majority of elite players assessed in this study had access to a registered nutritionist/dietitian.

To provide targeted education, sports nutrition professionals working with athletes need an understanding of the athletes’ current SNK and gaps in athletes’ SNK [32]. A systematic review assessing nutrition knowledge among athletes reported that common misconceptions include the roles of nutrients and their energy content [34]. Similar misconceptions were evident in the group of elite Gaelic footballers assessed in this study. Adequate energy intake is an important aspect of sport nutrition for athletes because sufficient energy is needed to protect the immune system, meet the needs of physical activity and enhance adaptation [12]. Gaelic footballers demonstrated a poor understanding regarding the energy density of macronutrients and effective weight-loss strategies, only 31% of Gaelic footballers in this study could identify that fat was the most energy-dense macronutrient. These misconceptions could help to explain the previous energy deficits observed in elite Gaelic footballers [9].

Previously reported CHO intakes of Gaelic football players also fall below team sport-specific recommendations [9,13,14,15]. In the present study, Gaelic footballers’ scores in the macronutrient section were average (54%); however, there was a lack of knowledge around the main source of energy for the exercising muscle, as protein was mistaken for this role rather than CHO by 31% of participants. This is in line with previous research which also suggests that athletes may have incorrect knowledge about the role of protein within the body [35]. In addition, 62% of participants seem to believe that CHO is the highest source of energy per gram, which may be a result of the low-CHO diet fads creating the misunderstanding that CHO is the main cause of weight gain [36]. The majority of Gaelic football players in the present study could accurately select foods that were high in protein, however, they demonstrated a poor understanding regarding CHO content of foods and CHO guidelines.

Athletes require an awareness of evidence-based sports nutrition guidelines to make appropriate dietary decisions, however, guidelines could be difficult for athletes to comprehend and remember [32]. In line with previous research, Gaelic footballers struggle to identify sports nutrition guidelines and dietary recommendations for CHO [32,37]. Only 44% of Gaelic footballers could identify that athletes should aim for a CHO intake of 6–10 g·kg^−1^ each training day to support 1–3 h of moderate to high-intensity exercise. Gaelic footballers demonstrated inadequate knowledge in relation to recommendations for foods to consume before and after competition. Only 50% of Gaelic football players could correctly identify that athletes should consume foods that are high in fluid and CHO pre-competition, while only 42% correctly identified that post-exercise nutrition should consist of high CHO and protein foods. Previous research has demonstrated that athletes do not follow sport nutrition guidelines and future research should focus on strategies to help Gaelic footballers remember and follow evidence-based sport nutrition recommendations [32,37,38].

A poor level of knowledge regarding nutritional supplements was reported, which is in line with previous research in team sport athletes [23,32]. Similar to previous research, the functions of supplements were poorly understood [23,32]. Concerningly, 11% of participants could not identify that nutrition supplements labels may contain false or misleading information, while 33% of participants did not realize that the purity and safety of supplements are not always tested. There is a real risk of inadvertent doping violations via the use of contaminated supplements [39]. Knowledge regarding nutritional supplements in elite Gaelic footballers could be improved and it is imperative that elite Gaelic footballers are made aware that due to contamination, there is a risk of violating an anti-doping rule associated with nutritional supplements.

Sub-group analysis indicated that there was no difference in nutrition knowledge across age groups, education levels, and divisional status. Previous studies have reported that level of education positively influences nutrition knowledge [32]; however, our data suggest that higher education level was not associated with better nutrition knowledge. Furthermore, in the current investigation, there were no significant differences in SNK between divisional statuses, which may be associated with the fact that the majority of players involved in the current study had regular access to a sports nutrition practitioner. There was no difference in SNK across age groups; investment in the area of sport nutrition is a relatively new phenomena in Gaelic football, so it is likely that older athletes have only recently been exposed to sports nutrition resources, therefore, do not have the benefit of years of experience in nutrition education.

Nutrition knowledge has a positive impact on dietary behaviours, and therefore it is important to explore the preferred nutrition information sources that Gaelic footballers use. This is the first study to investigate this topic in elite male Gaelic footballers. Gaelic footballers mostly obtained their nutritional information from reputable sources dietitian/nutritionists (84%) and athletic trainers/strength and conditioning coaches (70%), This finding is similar to professional Australian Rules football players who identified club dietitian, club trainer, and internet as their primary sources of nutrition information [28,30]. However, a large percentage still seek nutrition information from internet searches and social media, this is inadvisable as it can often be unreliable [40]. The SNK of practitioners working with elite Gaelic football squads was assessed during this study, each practitioner assessed provided nutritional advice to elite players in their squad. Despite, the small sample size, which is an obvious limitation, this is an area of research priority. This study highlights that team dietitians/nutritionists and athletic trainers/strength and conditioning coaches are players preferred sources of nutritional advice. Therefore, it is important that these individuals should be suitably qualified and have high levels of nutrition knowledge. SNK of practitioners assessed in this study was excellent with a mean score of 78.1%. The poor SNK of Gaelic footballers suggests that practitioners need to develop a better method of educating and translating sports nutrition principles to players.

The overall aim of nutrition education is to increase knowledge, support change and maintain appropriate dietary intake [19,41]. There is very little research on the optimal method to educate athletes, lectures and one-on-one counselling are the most common form of education intervention [19,42], but these can be impractical in many cases [41,43]. Nutritional support is usually offered on a part-time or consultancy basis, even in the professional environment, with limited time with the players [29]. It is unclear how often Gaelic footballers see the dietitian/nutritionist or how often they received advice. Elite Gaelic football squads normally meet two to five times a week, so time with the players is limited and greater importance is often placed on time spend with coaches and athletic trainers to work on physical, tactical and technical elements of the game.

Throughout conducting this research (which was part of a broader study on physical demands and nutritional requirements of Gaelic football), and through discussions with players and practitioners, we realized that the inadequate and irregular access to the sport nutritionists, and the limited time that the sports nutritionists spend with players might contribute to the poor SNK. We therefore propose that the practitioners may need to be innovative in their approach to translate evidence-based nutritional advice into practice. Online platforms such as mobile applications (apps) could provide sports nutrition practitioners the opportunity to make nutrition education interventions more time-efficient and successful [41,44]. Digital health interventions have demonstrated to be successful in enhancing nutrition knowledge and improving individuals’ behaviours which resulted in increased vegetable intake in young adults [45], and successful weight loss in overweight individuals [46]. Research suggests that athletes are receptive to a mobile device application as a nutrition resource [44]. Developing an appropriate nutritional strategy using online resources and mobile apps could help to improve nutritional knowledge and practice of Gaelic footballers. Future research is required to assess alternative education methods and delivery of nutrition information via web-based mobile apps.

The use of smartphone technology is becoming more prevalent and can provide a cost-effective method of maintaining better communication between athletes and practitioners especially when time with the athlete is limited [44]. Dunne et al. (2019) reported that sports nutritionists have adopted social media as part of the service they provide; 89% of the sports nutritionist interviewed used social media in their practice, with WhatsApp being the most frequently used. Social media platforms can facilitate mobile and visual learning, allowing communication with athletes across various environments and times of day to influence desired behaviors [44]. Athletes now want smaller more succinct levels of communication and content, therefore practitioners’ online delivery should be clear and concise, to effectively engage and influence the athlete [44]. Individuals are over six times more likely to retain information from an infographic compared to text alone, therefore, when possible practitioners should use visuals rather than text to increase engagement from athletes [44]. Sports nutritionists would benefit from incorporating online platforms and digital resource development as part of their service provision.

A limitation of this study is the low response rate of practitioners—18 were contacted, and only 8 agreed to participate and forward the questionnaire onto their squad of players. Furthermore, only 100 participants who started the questionnaire completed it. This may be due to the time-consuming nature of this particular questionnaire—it takes approximately 20 min to complete—or participants may have discontinued due to a perceived lack of knowledge. This could suggest that the cohort that completed the questionnaire are not representative of the wider population, but only a subsection with an interest in nutrition.

## 5. Conclusions

The results show that SNK among elite Gaelic footballers is poor. Practitioners working with Gaelic footballers should assess their understanding of evidence-based sport nutrition recommendations, so targeted education can be provided. It appears Gaelic footballers require nutrition education targeting energy and CHO requirements to improve dietary practices. Innovative education strategies need to be explored so improvements in knowledge can be made and translated to practice. Online platforms can be used to overcome physical and temporal barriers, and provide an inexpensive method to increase interaction, provide support and deliver information.

## Figures and Tables

**Table 1 nutrients-13-01081-t001:** Participant characteristics of Gaelic football players who completed the NSKQ.

Characteristics	Gaelic Footballers (*n* = 100)
Age	%
18–23	31
24–29	55
30+	14
Highest level of education	%
Secondary school	18
Third level non-degree (advanced certificates, completed apprenticeships, etc.)	8
Third level degree	40
Postgraduate degree	34
Divisional status	%
1	43
2	25
3	15
4	17

**Table 2 nutrients-13-01081-t002:** Mean sport nutrition knowledge scores of athletes (*n* = 100) and practitioners (*n* = 8) for each section of the nutrition knowledge questionnaire.

Section	Players Knowledge Score ± SD (%)	Practitioner Knowledge Score ± SD (%)
Weight management	54.4 ± 10.1	89.4 ± 9.4
Macronutrient	52.8 ± 10.0	81.3 ± 8.3
Micronutrient	34.2 ± 11.9	67.3 ± 12.0
Sports nutrition	42.3 ± 13.4	75.9 ± 9.8
Supplements	39.4 ± 14.0	79.2 ± 17.2
Alcohol	57.9 ± 10.0	67.2 ± 20.7
Total Nutrition Knowledge	47.6 ± 12.3	78.1 ± 8.3

NSKQ scoring; Poor (0–49%), Average (50–65%), Above average (66–75%), Excellent (75–100%).

**Table 3 nutrients-13-01081-t003:** Demographic characteristics and sports nutrition knowledge of elite Gaelic football players (*n* = 100).

Participant Characteristics	SNK Score ± SD (%)
Age	
18–23	46.5 ± 11.1
24–29	48.3 ± 13.2
30+	47.4 ± 11.6
Highest level of education	
Secondary school	44.6 ± 12.3
Third level non-degree	40.4 ± 9.3
Third level degree	47.9 ± 10.7
Postgraduate degree	50.4 ± 13.5
Divisional status	
1	45.0 ± 12.0
2	51.0 ± 11.00
3	44.9± 14.0
4	51.6 ± 11.2

No significant differences between groups.

**Table 4 nutrients-13-01081-t004:** Participants sources of information.

Sources of Information	*%*
Academic journal	12.0
Athletic trainer/strength and conditioning coaches	73.0
Coach	28.0
Doctor	18.0
Internet search	32.0
Mass media	13.0
Registered Dietitian/Nutritionists	84.0
Social media	37.0
Teammates	34.0

## Data Availability

The data presented in this study are available on request from the corresponding author.

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
