# Peer review of "Exploring Sports Nutrition Knowledge in Elite Gaelic Footballers"

_nutrients, 2021, doi:10.3390/nu13041081_

Round 1

Reviewer 1 Report

A few small typos/grammatical changes

Page 2 line 75 (include a in front of nutritionist/dietitian).

Line 80 Sports rather than sport nutrition

A nicely written paper that is interesting and warrants publication, however there are a few areas that could be strengthened prior to publication.

Introduction generally well done – a paragraph outlining some detail around links between nutrition knowledge and intake would be useful to demonstrate a sound understanding of this concept, additionally a discussion of types of knowledge and what is addressed in the questionnaire being used would also add to the paper. Eg. Declarative knowledge vs procedural knowledge. Ie does good nutrition knowledge always translate to better eating behaviours?

Table 3 – is this combined data for both practitioners and athletes? This isn’t clear. If it is both, it would be important to provide numbers for each of those sub-groups. It is a little confusing with some of the data relating to both groups and other just for the athletes. I’m not sure that the practitioner data really adds a lot given the small sample size. It is interesting that while the level of knowledge appears above average is was still lower than I would expect from a nutrition expert. Given the difference between knowledge of Trainers and sports nutritionists, these should possibly not be combined in the results.

Page 6 line 234 – “misunderstanding around main energy source” consider alternative wording for misunderstanding

Discussion – Tam, R., Beck, K., Manore, M., Gifford (nee Barnard), J., Flood, V., O'Connor, H. (2019). Effectiveness of Education Interventions Designed to Improve Nutrition Knowledge in Athletes: A Systematic Review. Sports Medicine, 49, 1769- 1786 might be a useful reference to consider when discussion appropriate nutrition interventions for athletes.

Smartphone and technology paragraphs in the discussion while interesting don’t really fit, there was no prior mention of technology in the paper and no specific results to really lead to this as a discussion point. Not sure that this is relevant to this particular paper without it being raised earlier. 

Author Response

Response to Reviewer 1:

A few small typos/grammatical changes

Point 1: Page 2 line 75 (include a in front of nutritionist/dietitian).

Response to Point 1: Thank you. Amended as recommended (The new Line Number is 105).

Point 2: Line 80 Sports rather than sport nutrition.

Response to Point 2: Thank you. Amended as recommended (The new Line Number is 110).

Point 3: A nicely written paper that is interesting and warrants publication, however there are a few areas that could be strengthened prior to publication.

Response to Point 3: Thank you for your positive comments. We are encouraged with your positive and productive feedback and do our best to address your comments and improve the quality to the best of our ability.

Point 4: Introduction generally well done – a paragraph outlining some detail around links between nutrition knowledge and intake would be useful to demonstrate a sound understanding of this concept, additionally a discussion of types of knowledge and what is addressed in the questionnaire being used would also add to the paper. Eg. Declarative knowledge vs procedural knowledge. Ie does good nutrition knowledge always translate to better eating behaviours?

Response to Point 4: Thank you. This is a very valid point. 

We have addressed the point through adding a few sentences to the introduction that provides more detail on the link between nutrition knowledge and dietary intake. We reference some papers that highlight that increasing nutrition knowledge through education interventions improves nutrition knowledge and corresponding dietary intake (Lines 66-74). Furthermore, we added another paragraph (Lines 80-101) in which we tried to strengthen the rationale in view of the previous studies and included comments on declarative and procedural knowledge.

Point 5: Table 3 – is this combined data for both practitioners and athletes? This isn’t clear. If it is both, it would be important to provide numbers for each of those sub-groups. It is a little confusing with some of the data relating to both groups and other just for the athletes. I’m not sure that the practitioner data really adds a lot given the small sample size. It is interesting that while the level of knowledge appears above average is was still lower than I would expect from a nutrition expert. Given the difference between knowledge of Trainers and sports nutritionists, these should possibly not be combined in the results.

Response to Point 5: Thanks for the productive feedback. 

The data in table 3 is just the athletes, and not both. We amended the table heading to reflect this (Lines 227-228).  We have also conceptually amended the narrative to clarify the matter.

In this particular issue, the feedback of the two reviewers are not fully aligned. While the second reviewer (legitimately) questions the values of the small sample, s/he advises the combination or preferably removal of the sections related to trainers and sport nutritionists. In order to address both reviewers' comments, first of all we removed the second part of the aim, and kept this only as an additional information within the article. We also provided clear explanations about the limitations of this small sample size (Lines 319-327). This way we hope that we have managed to address both reviewers’ comments.

Overall, Despite the small sample size, we feel the sports nutrition knowledge of trainers and sport nutritionists providing advice adds to the paper, especially considering that previous studies (e.g. Magee et al 2016, and Trakman et al 2016) recommended these as an area of the research priority. More importantly, and from practical perspectives, all these practitioners stated that they provide nutrition advice to players in their squad. Our study provides the first evidence on the source of nutrition knowledge for Gaelic football players, interestingly demonstrating that the team nutritionist, strength and conditioning coaches and social media are the top three sources of the knowledge. Hence, we feel that this is important that anyone providing nutrition advice should be suitably qualified and have high levels of nutrition knowledge, and as you are correctly identifying, this is not always the case, and within our small sample. We hope that our small sample and discussion provides a proof of concept and the need for more efficient methods of educating and engaging with athletes regarding sports nutrition. 

Point 6:  Page 6 line 234 – “misunderstanding around main energy source” consider alternative wording for misunderstanding.

Response to Point 6: Misunderstanding was amended to ‘lack of knowledge’ (The new Line Number is 266-7). 

Point 7: Discussion – Tam, R., Beck, K., Manore, M., Gifford (nee Barnard), J., Flood, V., O'Connor, H. (2019). Effectiveness of Education Interventions Designed to Improve Nutrition Knowledge in Athletes: A Systematic Review. Sports Medicine, 49, 1769- 1786 might be a useful reference to consider when discussion appropriate nutrition interventions for athletes.

Smartphone and technology paragraphs in the discussion while interesting don’t really fit, there was no prior mention of technology in the paper and no specific results to really lead to this as a discussion point. Not sure that this is relevant to this particular paper without it being raised earlier. 

Response to Point 7: This reference is now included in the introduction to highlight that education interventions can increase nutrition knowledge (Lines 66-68). It is also included in the discussion to discuss the aims of nutrition education intervention and strengthen the argument that face to face interventions are the most common type of nutrition education (Lines 330-3),

With regard to the smartphone and technology, you are right that these were not raised earlier and we therefore amended our wording to clarify that this was our opinion about the practical implications and potential direction for the future research (Lines 340-4).

Finally, we would like to take the opportunity to thank you for your productive comments and feedback.

Reviewer 2 Report

I am grateful for the opportunity of reviewing the article entitled “Exploring Sports Nutrition Knowledge in Elite Gaelic Footballers”, which presents results from a very interesting subject. However, there are some questions I would like authors to clarify before giving my approval for publication.

  • There is one more email address reported than authors are for this article, being the email alizado@hope.ac.uk (O.A.)
  • Abstract:
    • How could authors evaluate SNK among different divisional status when they previously state that the aim of this study was “…was to investigate the current level of SNK in elite Gaelic footballers…”. If, as later reported in the Results section, participant were enrolled in teams from division, 1, 2, 3, and 4, then they were not all elite athletes. If players from all these divisions are considered elite, then this should be further clarified by the authors.
  • Introduction:
    • First paragraph:
      • instead of reporting two mean ± SD values, I would suggest authors to report only a range (8160 – 9222 m)
      • Sentence starting with “The high-intensity nature of Gaelic football training…” should be checked, since it seems there is need to start with something like “Due to the high-intensity…”
    • Second paragraph:
      • it seems an “and” is missing in “The main focus of the competitive season is peak performance and recovery, the number and duration of training sessions…”
      • I do not understand why authors differentiate between “energy”, “carbohydrate”, and “protein”. Is it not carbohydrate an energy source?
      • Check for correct use of superscripts (i.e., g·kg-1 instead of g·kg-1)
      • Authors end this paragraph mentioning the lack of sufficient CHO intake in Gaelic footballers after competition, but no minimum threshold for this has been previously set within this paragraph.
    • Third paragraph:
      • With data/reference are authors using in order to support the claim that “…it is clear that Gaelic footballers have sub-optimal dietary intakes…”?
      • Based on the fact that the main aim of this work was “to evaluate the SNK of Gaelic footballers”, and, if as stated in this paragraph, “Previous research has identified that both elite and sub-elite Gaelic football players demonstrated poor nutrition knowledge”, what is the novelty of the current study? Authors should clarify this point.
    • Fourth paragraph:
      • Cite for Renard et al. (2020) do not follow format requirements from Nutrients. Please, check.
  • Methods:
    • Experimental design and participants:
      • Authors mention at the Introduction that the secondary aim of this work was “to investigate the knowledge of sport nutrition practitioners working with and advising Gaelic footballers and identify other sources of nutrition information”. However, they explain in this section that “Practitioners were also asked to complete the questionnaire themselves if they provided nutrition advice to players”. I would then argue that part of these practitioners would not be sport nutrition practitioners, and so the secondary aim is not precise.
    • Instruments
      • Information regarding number of questions is inconsistent. Numbers do not match, please double-check: “The questionnaire includes 89 questions across six sub-sections: 130 weight management (n =13), macronutrients (n =30), micronutrients (n =13), sports nutrition (n =13), supplementation (n =13) and alcohol (n =8)”
      • Authors explain that “All correct answers were totalled and resulted in the NSKQ score for each participant”, although later (Stats section) mention that “‘sport nutrition knowledge score’ refers to the proportion (%) of the correct answer”. Such information is inconsistent, and so it needs clarification.
    • Statistical analysis
      • Please, consider the information provided at https://www.ibm.com/support/pages/how-cite-ibm-spss-statistics-or-earlier-versions-spss in order to correctly cite SPSS developer.
      • Authors should explain what let them to create the specific age-groups generated for the analysis. Why these, and not different ones?
  • Results
    • Based on thresholds shown at Methods, 75% is not an SNK excellent value. Authors should better explained the thresholds used.
    • A sample size of 8 (i.e., for practitioners) leads to a very low statistical power for the statistical analysis, which would make difficult to present reliable conclusions.
    • Does table 3 refer to values from the entire sample, or just Gaelic football players? It is not clear. Same comment applies to data reported in table 4.
  • General comments:
    • There is some incongruency between the title of the article, its objectives, and its conclusions, since the title only refers to “Elite Gaelic Footballers”, which is in line with its main objective, but does not have nothing to do with the secondary aim “to investigate the knowledge of sport nutrition practitioners working with and advising Gaelic footballers”. I suggest authors to try to follow the same line of argumentation from the title to the conclusions.
    • Moreover, the secondary aim refers to “sport nutrition practitioners”, whereas later in the text it seems this subsample is not just compound by sport nutrition practitioners, but by “six athletic trainers/strength and conditioning coaches who received some nutrition education during their sports science-related degrees and two nutrition practitioners“. This might mislead readers, and so authors should clarify this point along the article.
    • Authors discuss data which is not previously reported within the results section. For instance, authors claim that “it is more appropriate for nutritionists to provide nutritional advice to players” based on a separate analysis for athletic trainers (76.4 %) and sports nutritionists (83.0 %) scoring regarding SNK. Firstly, given that only two nutrition practitioners were tested, it is highly probable that these results are underpowered, meaning that authors conclusion is potentially incorrect.

Best Regards.

Author Response

Response to Reviewer 2:

I am grateful for the opportunity of reviewing the article entitled “Exploring Sports Nutrition Knowledge in Elite Gaelic Footballers”, which presents results from a very interesting subject. However, there are some questions I would like authors to clarify before giving my approval for publication.

Point 1: There is one more email address reported than authors are for this article, being the email alizado@hope.ac.uk (O.A.)

Response to Point 1: Thank you for this point and apology for this error. We have clarified through removing the unnecessary email.

Point 2: Abstract:

  1. How could authors evaluate SNK among different divisional status when they previously state that the aim of this study was “…was to investigate the current level of SNK in elite Gaelic footballers…”. If, as later reported in the Results section, participant were enrolled in teams from division, 1, 2, 3, and 4, then they were not all elite athletes. If players from all these divisions are considered elite, then this should be further clarified by the authors.

Response to Point 2: Thank you. Clarification was added as advised (Lines 134-135).

We agree with your comments about the complexities and controversies in defining the term elite athletes, as seen within the commentary of ‘What does ‘elite’ mean in sport and why does it matter?’ available from https://www.bases.org.uk/imgs/51_article_p6527.pdf

Gaelic football is a unique sport. It is an amateur sport, and it is parochial in nature. Players play for their local club side (sub elite) and the best Gaelic footballers in each of the 32 counties in Ireland are selected to represent their county team (elite) who compete in the All-Ireland Championship and the National League competitions. The All-Ireland Championship competition is the most prestigious competition in Gaelic football, and it is played by all county teams in a knockout cup format. It attracts mass national interest with over 80,000 spectators present at the All-Ireland final. In the National Football League competition, the second most prestigious competition, the teams are divided into four ranked divisions with 8 teams in each division, with promotion and relegation between divisions. Each player assessed in this study plays at the highest level of the game (The All-Ireland Football Championship). It is not possible for a player to play at a higher level than their county team, this is why we use the term elite. Previous research (Mangan et al., 2017; McGahan, 2018), have compared the physical demands of Gaelic football match-play across divisions yet still refer to all participants as elite despite their divisional status as it is the highest level of the sport that they can compete in.

Point 3: Introduction:

  1. First paragraph:
    1. instead of reporting two mean ± SD values, I would suggest authors to report only a range (8160 – 9222 m)
    2. Sentence starting with “The high-intensity nature of Gaelic football training…” should be checked, since it seems there is need to start with something like “Due to the high-intensity…”

Response to Point 3: Thank you.

The text has been amended as recommended (Lines 31-32 and 37-8).

Point 4: Introduction:

  1. Second paragraph:
    1. it seems an “and” is missing in “The main focus of the competitive season is peak performance and recovery, the number and duration of training sessions…”
    2. I do not understand why authors differentiate between “energy”, “carbohydrate”, and “protein”. Is it not carbohydrate an energy source?
    3. Check for correct use of superscripts (i.e., g·kg-1 instead of g·kg-1)
    4. Authors end this paragraph mentioning the lack of sufficient CHO intake in Gaelic footballers after competition, but no minimum threshold for this has been previously set within this paragraph.

Response to Point 4: Thank you.

  1. We amended the text as advised (Line 44).
  2. You are absolutely correct. We have distinctively mentioned energy, carbohydrate and protein in our sentence first of all to be loyal to the reference that we are referring to, secondly, because of particular need for the prioritisation recommendations for these macronutrients (in comparison with fat, which is also a macronutrients but perhaps carrying less emphasis in this context), and thirdly in view of the broader work that we are publishing in the same journal simultaneously, on energy and carbohydrate needs of Gaelic Football players. 

We hope that the references to carbohydrates requirements coming later in the same paragraph helps to explain the role of carbohydrates in particular as the energy source.

iii. We amended the superscripts (Lines 51-60). Thank you. 

  1. The paragraph amended for clarification. Lines 48-50 were added to clarify the context. Lack of sufficient CHO intake after competition (reporting finding of another study) was removed to avoid confusion.

Point 5: Introduction:

  1. Third paragraph:
    1. With data/reference are authors using in order to support the claim that “…it is clear that Gaelic footballers have sub-optimal dietary intakes…”?
    2. Based on the fact that the main aim of this work was “to evaluate the SNK of Gaelic footballers”, and, if as stated in this paragraph, “Previous research has identified that both elite and sub-elite Gaelic football players demonstrated poor nutrition knowledge”, what is the novelty of the current study? Authors should clarify this point.

Response to Point 5: Thank you.

  1. References were added as advised and the sentence was amended to clarify (Lines 63-65).
  2. Thank you for this excellent point. A full paragraph was added to clarify and strengthen the rationale (Lines 80-101).

In addition, the current study provides the first evidence of the sources of nutrition information and knowledge for Gaelic football players.

Point 6: Introduction:

  1. Fourth paragraph:
    1. Cite for Renard et al. (2020) do not follow format requirements from Nutrients. Please, check.

Response to Point 6: Thank you.

  1. The format of reference was amended as advised (Line 109).

Point 7: Methods:

  1. Experimental design and participants:
    1. Authors mention at the Introduction that the secondary aim of this work was “to investigate the knowledge of sport nutrition practitioners working with and advising Gaelic footballers and identify other sources of nutrition information”. However, they explain in this section that “Practitioners were also asked to complete the questionnaire themselves if they provided nutrition advice to players”. I would then argue that part of these practitioners would not be sport nutrition practitioners, and so the secondary aim is not precise.

Response to Point 7: Thank you for this very valid point. 

In this particular issue, the feedback of the two reviewers are not fully aligned. While the first reviewer (legitimately) questions the small sample, s/he advises to keep this. In order to address both reviewers' comments, first of all we removed the second part of the aim, and kept this only as an additional information within the article. We also provided clear explanations about the limitations of this small sample size and we also did not differentiate between nutritionists and athletic trainers. We now refer to them as practitioners who provide nutritional advice, which is the case (319-27). This way we hope that we have managed to address both reviewers’ comments.

Overall, Despite the small sample size, we feel the sports nutrition knowledge of trainers and sport nutritionists providing advice adds to the paper, especially considering that previous studies (e.g. Magee et al 2016, and Trakman et al 2016) recommended these as an area of the research priority. More importantly, and from practical perspectives, all these practitioners stated that they provide nutrition advice to players in their squad. Our study provides the first evidence on the source of nutrition knowledge for gaelic football players, interestingly demonstrating that the team nutritionist, strength and conditioning coaches and social media are the top three sources of the knowledge. Hence, we feel that this is important that anyone providing nutrition advice should be suitably qualified and have high levels of nutrition knowledge, and as you are correctly identifying, this is not always the case, and within our small sample. We hope that our small sample and discussion provides a proof of concept and the need for more efficient methods of educating and engaging with athletes regarding sports nutrition. 

Point 8: Methods:

  1. Instruments
    1. Information regarding number of questions is inconsistent. Numbers do not match, please double-check: “The questionnaire includes 89 questions across six sub-sections: 130 weight management (n =13), macronutrients (n =30), micronutrients (n =13), sports nutrition (n =13), supplementation (n =13) and alcohol (n =8)”
    2. Authors explain that “All correct answers were totalled and resulted in the NSKQ score for each participant”, although later (Stats section) mention that “‘sport nutrition knowledge score’ refers to the proportion (%) of the correct answer”. Such information is inconsistent, and so it needs clarification.

Response to Point 8: Thank you.

  1. The number of questions in the section for supplementation is 12. We do apologise for the typo and confusion. This has been amended (Line 162). 
  2. The wording of the statements were amended to clarify the confusion (Lines 165-167). 

Point 9: Methods

  1. Statistical analysis
    1. Please, consider the information provided at https://www.ibm.com/support/pages/how-cite-ibm-spss-statistics-or-earlier-versions-spss in order to correctly cite SPSS developer.
    2. Authors should explain what let them to create the specific age-groups generated for the analysis. Why these, and not different ones?

Response to Point 9: Thank you. 

  1. Reference to SPSS was amended as ‘IBM Corp. Released 2016. IBM SPSS Statistics for Windows, Version 24.0. Armonk, NY: IBM Corp’ as advised (Lines 181-183).
  2. A sentence was added to the participant characteristics after the mean age statement to clarify the groups (Lines 195-197).

Point 10: Results

  1. Based on thresholds shown at Methods, 75% is not an SNK excellent value. Authors should better explained the thresholds used.
  2. A sample size of 8 (i.e., for practitioners) leads to a very low statistical power for the statistical analysis, which would make difficult to present reliable conclusions.

iii. Does table 3 refer to values from the entire sample, or just Gaelic football players? It is not clear. Same comment applies to data reported in table 4.

Response to Point 10: Thank you. 

  1. Clarification was provided to explain the threshold (Lines 173-174).
  2. Please see response to Point 7.

iii. The data in table 3 is just the athletes, and not both. We amended the table heading to reflect this (Lines 227-228). 

Point 11: General comments:

  1. There is some incongruency between the title of the article, its objectives, and its conclusions, since the title only refers to “Elite Gaelic Footballers”, which is in line with its main objective, but does not have nothing to do with the secondary aim “to investigate the knowledge of sport nutrition practitioners working with and advising Gaelic footballers”. I suggest authors to try to follow the same line of argumentation from the title to the conclusions.
  2. Moreover, the secondary aim refers to “sport nutrition practitioners”, whereas later in the text it seems this subsample is not just compound by sport nutrition practitioners, but by “six athletic trainers/strength and conditioning coaches who received some nutrition education during their sports science-related degrees and two nutrition practitioners“. This might mislead readers, and so authors should clarify this point along the article.

iii. Authors discuss data which is not previously reported within the results section. For instance, authors claim that “it is more appropriate for nutritionists to provide nutritional advice to players” based on a separate analysis for athletic trainers (76.4 %) and sports nutritionists (83.0 %) scoring regarding SNK. Firstly, given that only two nutrition practitioners were tested, it is highly probable that these results are underpowered, meaning that authors conclusion is potentially incorrect.

Response to Point 11: Thank you.

  1. Thank you. We believe that we have amended this and provided clarification. Please also see response to Point 7.
  2. Thank you. The secondary aim of the article was removed to avoid confusion.

iii.         The section was removed to avoid confusion.

Finally, we would like to take the opportunity to thank you for your productive comments and feedback.
